

# Design and Evaluation of an Efficient High-Precision Ocean Surface Wave Model with a Multiscale Grid System (MSG_Wav1.0)

Jiangyu Li[1], Shaoqing Zhang[*1,2], Qingxiang Liu[3], Xiaolin Yu[1,2], Zhiwei Zhang[1,2]

[1]Frontier Science Center for Deep Ocean Multispheres and Earth System (FDOMES) and Physical Oceanography
Laboratory, Ocean University of China, Qingdao, 266100, China.
[2]Qingdao National Laboratory for Marine Science and Technology, Qingdao, 266100, China.
[3]College of Oceanic and Atmospheric Sciences, Ocean University of China, Qingdao, 266100, China.

*Correspondence to*: Shaoqing Zhang (szhang@ouc.edu.cn)

**Abstract.** Ocean surface waves induced by wind forcing and topographic effects are a crucial physical process at the air-sea
interface, which significantly affect typhoon development, ocean mixing, etc. Higher-resolution wave modeling can simulate
more accurate wave states, but requires huge computational resources, making it difficult for Earth system models to include
ocean waves as a fast-response physical process. Given that high-resolution Earth system models are in demand, efficient
high-precision wave simulation is necessary and urgent. Based on the wave dispersion relation, we design a new wave
modeling framework using a multiscale grid system. It has the fewest number of fine grids and reasonable grid spacing in
deep water areas. We compare the performance of wave simulation using different spatial propagation schemes, reveal the
different reasons for wave simulation differences in the westerly zone and the active tropical cyclone region, and quantify the
matching of spatial resolutions between wave models and wind forcing. A series of numerical experiments show that this
new modeling framework can more precisely simulate wave states in shallow water areas without losing accuracy in the
deep ocean while costing a small fraction of traditional simulations with uniform fine-gridding space. With affordable
computational expenses, the new ocean surface wave modeling can be implemented into high-resolution Earth system
models, which may significantly improve the simulation of the atmospheric planetary boundary layer and upper-ocean
mixing.

## 1 Introduction

Ocean surface waves induced by wind forcing and topographic effects significantly affect the flux exchange at the air-sea
interface (e.g., Garg et al., 2018; Qiao et al., 2010; Sullivan and McWilliams, 2010). Ocean surface waves can modify the
intensity and structure of tropical cyclones by sea surface roughness and ocean spray (e.g., Bao et al., 2000; Zhang et al.,
2021). It can also mitigate the overestimated sea surface temperature in summer in ocean circulation models by enhancing
ocean mixing with the help of wave breaking, wave-turbulence interaction, and Langmuir circulation (e.g., Hughes et al.,
2021; Zhang et al., 2012). Besides, ocean surface waves have a contribution to the transport of sea surface floating litter
(Higgins et al., 2020) and underwater spilled oil (Cao et al., 2021) because there are Stokes drifts as ocean surface waves





propagate forward. Furthermore, driven by strong wind, disastrous waves with extreme wave heights (Wu et al., 2021) can cause huge economic losses and serious casualties to coastal residents (Tao et al., 2018). Therefore, obtaining the accurate distribution of wave states in time and space is extremely necessary to study atmospheric and oceanic phenomena and then guide human production and life.

Because of their small scales with wavelengths ranging from centimeters to hundreds of meters, ocean surface waves are difficult to be resolved explicitly in large-scale numerical models (Brus et al., 2021). Phase-averaged wave models only describe the statistical characteristics of wave states in every fluid unit, which is dominated by source-sink terms (e.g., WAMDI group, 1988; Yang et al., 2005). Up to now, several studies have been done to enhance wave simulation accuracy, such as choosing the appropriate parameterization schemes for different external forcings (e.g., Kaiser et al., 2022; Stopa et

al., 2016), optimizing the parameterizations of source-sink terms (e.g., Liu et al., 2019; Zieger et al., 2015), and implementing more physical processes (e.g., Mentaschi et al., 2015; Rogers and Holland, 2009).

A higher-resolution model consisting of finer grid units can better resolve complex topographic features and meandering shorelines (e.g., Chawla and Tolman, 2008; Tolman, 2003). Wave models with higher resolution can better express the blocking effect of small islands and take into account more local responses to high-precision environmental forcings,

especially wind forcing. Thus, enhancing wave model resolution also is a feasible way to obtain high-precision wave states. However, high-resolution simulation in the whole domain could be very expensive, which is limited by the computational resources available. It is inconvenient for high-precision operational wave forecasting and also blocks ocean surface waves from participating in high-resolution Earth system models as a fast-response physical process at the air-sea interface (e.g., Dunne et al., 2020; Jungclaus et al., 2022). Usually, one uses the coarse-resolution simulation result as an approximation if

coupled systems consider ocean surface waves (Bao et al., 2020; Danabasoglu et al., 2020).

Due to the advancement of high-performance computing (HPC), high-precision operational wave forecasting and high-resolution Earth system models are in demand, which need high-precision ocean surface wave modeling with high efficiency urgently. After analyzing the theory that wave modeling describes the average characteristics of wave states using the wave action density spectrum as a statistical variable, regulated by the wave dispersion relation, we design a new wave modeling

framework based on a multiscale grid system. Then we compare the performance of this system using four numerical schemes in geographical space, reveal the different reasons for wave simulation differences in two areas of strong wind, and quantify the matching of wave model grid resolution and wind signal. The optimized multiscale grid system is much finer in coastal areas, but with a reasonable coarse grid spacing in open oceans. It can eliminate the excessive usage of computational resources due to double calculations in the overlapping area and the two-way information exchange, and therefore is more

convenient to exchange fluxes with the atmosphere and ocean in complex Earth system models, compared with traditional multi-layer nesting.

This paper is organized as follows. Section 2 displays the importance and constraint of high-resolution wave simulation and analyzes the feasibility of efficient and high-precision wave modeling based on theoretical analysis and numerical simulation of the wave dispersion relation. Section 3 designs a new wave modeling framework with the unstructured triangular



multiscale grid system after the comparison of different multiscale grid systems. Section 4 systematically tests and thoroughly evaluates the performance of this new modeling in deep and shallow water areas using a series of numerical experiments. Finally, section 5 gives some summary and discussions.

## 2 Raising the scientific idea

### 2.1 The importance and constraint of high-resolution wave modeling

In this section, we will first analyze the characteristic of wave simulation using traditional structured grids (or regular latitude-longitude grids) with different model resolutions by a set of experiments shown in Table 1. The design of these experiments is briefly introduced below. All physical processes in the wave model WaveWatch III version 5.16 (hereafter WW3; Tolman, 1991) are activated, of which parameterization settings can refer to Li and Zhang (2020). The needed wind forcing is from the reanalysis dataset ERA5 of the European Center for Medium-Range Weather Forecasts (ECMWF), with

a spatial and temporal resolution of 0.25° and 6 hours, respectively. The shoreline data can be obtained from the Global Self-consistent Hierarchical High-resolution Shoreline (GSHHS) dataset, National Oceanic and Atmospheric Administration (NOAA). The topography data is from the NOAA ETOPO1 dataset with a spatial resolution of 1′. For simplicity, we choose the Asia-Pacific area (39°E-178.5°E, 16°S-62.5°N) to explain our scientific idea. The required wave boundary information is from the global wave simulation (0°-359°, 75°S-75°N) using a traditional structured grid with 1° resolution driven by ERA5

wind.

Driven by the same ERA5 wind, Figure 1 shows the spatial distributions of significant wave heights (hereafter SWHs) around Taiwan Island, China (119°E-123°E, 21°N-26°N) in January 2018. They are from wave simulations (briefly as "WS") using a traditional structured grid system (briefly as "s" in the superscript) with 1°, 0.5°, 0.25°, and 0.125° model resolutions (denoted as the subscript), called $WS_1^s$, $WS_{0.5}^s$, $WS_{0.25}^s$, and $WS_{0.125}^s$ in Tab. 1. The ability to identify land and ocean in wave

models is a prerequisite to obtain accurate wave states. However, there is an obvious mismatch between the real (surrounded by black lines, from the GSHHS dataset) and identified (white fill) locations of Taiwan Island and the Chinese mainland, particularly in $WS_1^s$ and $WS_{0.5}^s$ (Figs. 1a and 1b). The lack of representation of some islands is a major local error source (Tolman, 2003). When model resolutions are coarse (Figs. 1a and 1b), the blocking effects of the Penghu Islands (for example) are not well-expressed. Unsurprisingly, as model resolutions increase in $WS_{0.25}^s$ and $WS_{0.125}^s$ (Figs. 1c and 1d), the

above poor shoreline fitting and island representativeness are improved. Nevertheless, even if the model resolution is increased to 0.125° in Fig. 1d, there is still a gap between the real and identified shorelines and topography. For instance, Green Island is too small to be resolved in wave models, which will be approximated with obstruction grids (Chawla and Tolman, 2008) or parameterized with a source term (Mentaschi et al., 2015) instead.

It is generally believed that the finer the model resolution is, the more accurate wave states can be obtained. Since we don't

have real wave states in the whole domain, simulation results obtained from the experiment using the structured grid with 0.0625° resolutions (named $WS_{0.0625}^s$ in Tab. 1) are considered as a reference to verify the influence of different model





resolutions on wave simulation accuracy. The linear interpolation method is used to calculate the SWH root mean square differences (hereafter RMSDs). Figure 2 shows the spatial distributions of SWH RMSDs around the Asia-Pacific area in January 2018. The simulated RMSDs are smaller as the model resolution gets finer. When the model resolution is 1°, 0.5°,

0.25°, and 0.125° (Figs. 2a-2d), the corresponding RMSD is 0.11, 0.07, 0.04, and 0.02 m, respectively.

Figure 3 shows the time consumption of the above simulation experiments using a structured grid system under the same computational condition. When the model resolution is coarse ($WS_1^s$, $WS_{0.5}^s$, $WS_{0.25}^s$, and $WS_{0.125}^s$), the consumed time is acceptable for us. However, when the model resolution is improved from 0.125° to 0.0625°, the consumed time increases from 1.92 to 33.79 hours dramatically, most likely due to the WW3 parallelism called card deck and a large amount of model

data output. The common approach to shortening computational time is to add parallel computing cores if computational resources are abundant. It is feasible when the cores used are smaller than a certain threshold. While as the number of cores increases, the saved computational time can be offset by the increased time from the excessive information exchange between the cores (Feng et al., 2016). Not to mention, when computational resources are limited, it is impossible to achieve high-resolution wave simulation. In the future, if higher-resolution, longer-duration, and larger-area wave states are needed,

it will take huge computational resources and time, even as expensive as the atmosphere-ocean coupled models (Brus et al., 2021).

In summary, higher-resolution wave models have better ability in shoreline fitting and topography description (Fig. 1) and can simulate more precise wave states (Fig. 2). However, high-resolution wave simulation with a uniform fine-gridding space requires huge computational resources (Fig. 3), which is a challenge to high-precision operational forecasting systems

and high-resolution Earth system models. Therefore, efficient and high-precision wave modeling is very necessary and urgent.

## 2.2 Analysis and understanding of the wave dispersion relation

As we know, wave modeling is regulated by the wave dispersion relation, here we will reintroduce it. The dispersion relation, a relationship between relative frequency ($\sigma$), wave number ($k$), and water depth ($d$), represents the nature and

characteristics of ocean surface waves. It is expressed by $\sigma^2 = gktanh(kd)$, where $g$ and $tanh$ are the gravitational acceleration and hyperbolic tangent function, respectively. In classical ocean surface wave theory, the magnitude relationship of $\frac{d}{l} > \frac{1}{2}$, $\frac{1}{20} < \frac{d}{l} \leq \frac{1}{2}$, and $\frac{d}{l} \leq \frac{1}{20}$ is used to determine deep, intermediate, and shallow water areas, where $l$ represents the wavelength. After a simple mathematical limit operation, the wave dispersion relation $\sigma^2 = gktanh(kd)$ is simplified to $\sigma^2 = gdk^2$ and $\sigma^2 = gk$ in shallow ($\frac{d}{l} \leq \frac{1}{20}$) and deep ($\frac{d}{l} > \frac{1}{2}$) water areas, respectively.

To more vividly show the meaning of the wave dispersion relation, a schematic diagram of wave propagation characteristics described in different water areas and simulated with different spatial resolutions is shown in Figure 4. In deep water areas, ocean surface waves have large wavelengths and long wave periods. Because they are insensitive to topographic features (represented by water depth $d$ in the above dispersion relation), wave models with coarse or fine resolution, consisting of





coarse or fine grid units, have good performance in simulating wave states without losing accurate responses to wind signals.

When ocean surface waves travel from deep to intermediate water areas (their boundary is marked with a green vertical bar), the wavelength decreases, and the wave height increases. The effects of topographic features (thick black line) on the wave states are activated. These features (such as sea peaks and valleys) are well-represented/excessively-smoothed using fine/coarse resolution models (thick red/blue lines), which directly affects wave simulation accuracy. Moreover, when ocean surface waves reach coastal areas with very shallow water, more complex physical processes should be considered, such as depth-induced wave breaking, wave scattering and reflection, and so on. However, the described topographic features are distorted even when using fine-resolution models, let alone coarse-resolution models. This situation will lead to very poor simulation precision (as shown in Fig. 1d). Thus, wave model resolution needs to be improved constantly, especially in coastal areas. It's worth mentioning that this figure is a schematic diagram and does not represent the actual wave modeling process (using wave action density spectrum as the integral variable) and spatial scales of ocean surface waves, only to illustrate our idea.

Next, we will use numerical simulation results to further understand the above theoretical characteristic. Figure 5 shows the evolution of SWH differences (representing errors) around the South China Sea (105°E-125°E, 0°N-27°N) on the first day of model integration. The wave states are resting at the first moment of the model run (00:00 UTC, November 1, 2017). After that, ocean surface waves begin to generate and propagate, induced by wind forcing and topographic effects. Driven by strong wind, ocean waves in the northwest South China Sea have rapid responses at the 1st integral time step (00:15 UTC, November 1, 2017). Because coarse-resolution models lack representation of complex topography ($WS_1^s$ for example), SWH differences are generated at the beginning of the model run, especially in the central part of the South China Sea where the water depth gradient is very large (northeastern-southwestern direction). They are propagated along the wave motion way, which can be observed clearly at the 4th integral time step in Fig. 5a. As time passes, the simulated differences are constantly generated and propagated to the deep ocean, driven by the strong wind (Figs. 5e and 5i). At the 24th hour of model integration, they are almost distributed over the whole South China Sea (Fig. 5m). At the same time, driven by weak wind, SWH differences are small and their effects on the surrounding sea areas are weak relatively in the southeast South China Sea (the first column). As we expected, with the increase of model resolution, there is a higher representation of topographic features, so the simulated differences gradually decrease. They are almost imperceptible when the model resolution is 0.125° ($WS_{0.125}^s$, the fourth column). Please see Zhongsha Islands circled by dashed boxes in the first column and last row for a more intuitive observation.

## 2.3 On the feasibility of efficiently modeling ocean surface waves

Based on the above theoretical analysis and numerical simulation, we have the following understanding. (1) In shallow and intermediate water areas, wave states are very sensitive to topographic features, especially in coastal areas. Therefore, a finer-resolution wave model consisting of smaller fluid units is necessary to better describe the complex topographic features and meandering shorelines. This way can reduce wave simulation errors in shallow water areas and weaken their effects on



the surrounding sea areas. It also takes into account more local responses driven by high-precision environmental forcings, especially wind forcing. (2) In deep water areas, wave states are insensitive to topographic effects. Then, a coarse-resolution model is suggested to save computational resources without sacrificing accurate responses to external forcings.

Therefore, similar to the classical wave theory, we choose the magnitude relationship between $\frac{d}{l}$ and $\frac{1}{2}$ to determine shallow $(\frac{d}{l} \leq \frac{1}{2})$ and deep $(\frac{d}{l} > \frac{1}{2})$ water areas for simplification. Here, the "shallow" water areas are a general notion, including the shallow and intermediate water areas defined in classical theory, where topographic effects should be taken into account in wave simulation. It's important to note that we only follow the idea of dividing different water areas from the classical theory, and do not change the expression of the wave dispersion relation in all numerical simulation experiments. Previous

studies have used a specific/gravity water depth as a criterion to classify different waters (e.g., Brus et al. 2021; Li, 2012; Mao et al., 2015), which has achieved good results in saving wave simulation time. The method used in this paper is a direct application of the wave dispersion relation, then can minimize the number of fine grids. This will further improve wave simulation efficiency, which is very much needed for the Earth system models considering the ocean surface wave process.

Therefore, we can design a new wave modeling framework with a multiscale grid system much finer in coastal areas but

relatively coarse in open oceans, to achieve efficient and high-precision wave simulation. This wave modeling idea is feasible preliminarily since the global ocean is almost covered by deep water with only a small portion of shallow water, such as only 2.7% of shallow water in the Asia-Pacific area. Next, we will introduce the different implementations of building this framework, the factors to consider for designing a multiscale grid system, and the performance of this framework in detail.

## 3 Design of an efficient and high-precision wave modeling framework

### 3.1 Multiscale grid systems

Multiscale grid systems are usually made up of multiple polygons with different spatial sizes. Now, two multiscale grid systems are available in wave models. One is made up of rectangles with different sizes (Li, 2011), named unstructured rectangular multiscale grid in this paper. And the other is made up of triangles (e.g., Roland et al., 2009; Zijlema, 2010),

called unstructured triangular multiscale grid ("utms" for short, superscripts of experiment names in Tab. 1). They have similar design ideas, setting fine-resolution meshes in shallow water areas to enhance simulation accuracy, and coarse-resolution meshes in deep water areas to save computation resources. At the same time, to avoid a sharp change in coastal water depth, setting modest-resolution meshes in transitional water areas ensures a stable calculation. Note that the transitional water areas here are a part of deep water areas, which are different from the intermediate water areas in the

classical wave theory. Now, using simple diagrams in Figure 6, the generation steps of these two grids both with variable resolutions from Δx in shallow water areas to 2Δx in transitional water areas and then to 4Δx in deep water areas, and their





performance are briefly introduced. Note that curvilinear grids as an extension of traditional structured grids (Rogers and Campbell, 2009) are not discussed here.

### 3.1.1 Generation of multiscale grid systems

Steps for making unstructured rectangular multiscale grid systems are described as follows (Hou et al. 2022). The study area can be divided into 2×2 rectangular groups with 4Δx resolutions. Looping for every group, if there is no land inside, the group is marked with blue lines in Fig. 6a. Otherwise, the group can be further divided into 2×2 boxes with 2Δx resolutions. Similarly, looping for every box, it is marked with magenta lines if the box is covered with water everywhere. Or the box is divided into 2×2 cells with Δx resolution. Cells near shorelines can be identified as land or ocean by judging the land-ocean

ratio in every cell. The actual and fitted shorelines are marked with thick black and red lines, respectively. Now, the unstructured rectangular multiscale grid is generated. Note that the scale of two adjacent meshes is 1:1 or 1:2.

The steps of generating an unstructured triangular multiscale grid are described in the following. In the beginning, obtaining fine shoreline data is necessary. Next, with the help of shorelines and two types of control lines marked with thick red, magenta, and blue lines in Fig. 6b, the spatial resolution in shallow, transitional, and deep water areas can be set to Δx, 2Δx,

and 4Δx, respectively. Once reasonable control lines are ready, a lot of triangles with different spatial sizes are generated quickly. Now, making the unstructured triangular multiscale grid is finished. Note that if the grid resolution is set to 4Δx, this does not mean that the length of three elements in every triangle is 4Δx exactly, but varies within a reasonable range around 4Δx.

### 3.1.2 Comparison of two grid systems

Here will further compare the performance of wave simulation using different grid systems with the same fine resolution. The lower panels of Fig. 6 show spatial distributions of SWHs from wave simulation using the traditional structured and unstructured triangular grids both with 0.125° resolutions (named $WS_{0.125}^{s}$ and $WS_{0.125}^{ut}$ in Tab. 1). It's like using the finest spatial resolution (Δx) throughout the whole domain in the upper panels (Figs. 6a and 6b). Compared to those using the structured grid (red lines in Fig. 6c), wave models using the unstructured grid (red lines in Fig. 6d) have a better ability to fit

the actual land-ocean shorelines (black lines in Figs. 6c and 6d). This is the reason why the latter has simulation results at all 9 available Chinese oceanic stations (Table 2), while the former has simulation data only at 4 stations, including XCS, NJI, BSG, and DCN, respectively. Since wave simulation using different grids performs similarly at these four stations, the results at station BSG are used here as an example to illustrate. This station marked with yellow stars in Figs. 6c and 6d lies near a group of small islands (a distance from the mainland), which are not enough to be resolved in wave models using

structured or unstructured grids with 0.125° resolutions. The former uses sub-grid obstacles with different levels of transparency for approximation, while the latter directly treats them as water areas. When waves travel from the open ocean to the mainland in a southeast direction, ocean surface waves at this observation station behind these islands are





underestimated resulting from a lot of wave energy dissipation caused by excessive blocking in wave models using a structured grid. For example, the observed average SWH is 1.28 m at the valid observed time in July 2018, and the simulated

SWHs are 1 m and 1.23 m in $WS_{0.125}^{s}$ and $WS_{0.125}^{ut}$ (Figs. 6c and 6d), respectively. Therefore, wave models using the unstructured triangular grid have more advantages than those using the traditional structured grid in shoreline fitting and coastal simulation accuracy, while they take almost the same computational time (2.04 and 1.92 hours in the following Figure 13).

### 3.2 Design of a new wave modeling framework

Considering the advantages of triangular grids in coastal areas (e.g., Engwirda, 2017; Roberts et al., 2019) and the follow-up sustainability of this work, we design the first version of a new wave modeling framework using an unstructured triangular multiscale grid to achieve the goal of efficient and high-precision wave simulation. The generated steps in Surface-water Modeling System software (SMS) are described as follows. Similar to previous papers, we will empirically set the spatial resolution of this multiscale grid in different water areas. In the following section, we will show the performance of this grid

setting and optimize it further, along with some tips for designing the grid resolution, particularly in deep water areas, which is friendly for readers to follow.

Step 1: obtaining and optimizing shorelines. Theoretically, with the support of high-resolution topography and shoreline datasets, mesh resolution can be refined infinitely (e.g., Li and Saulter, 2014) in shallow water areas to simulate higher-precision wave states. Fine shoreline data comes from the NOAA GSHHS dataset with a 1 km resolution, and topography

data comes from the NOAA ETOPO1 dataset with 1′ resolution. In practice, trading off the simulation accuracy and computational resource consumption, we set shoreline resolution to 0.125° (red lines in Figure 7) for a preliminary test. Proper shoreline adjustment is suggested if there is any unsuitability, which is very important to accurately obtain coastal wave states (Fig. 6d). When finer shoreline data is available in key areas, the shorelines should be further refined if necessary.

Step 2: setting control lines with different spatial resolutions. As stated in section 2.2, wave states are insensitive to topographic features in deep water areas, which can be simulated using coarse-resolution models. Here, we determine the boundary locations between shallow and deep water areas based on the relationship between the water depth and half of the minimum mean wavelength. These two variables are derived from wave simulation results with a resolution of 0.0625° ($WS_{0.0625}^{s}$ in Tab. 1) in 2018. Then, the control lines following this boundary can be set to 0.5° (magenta lines in Fig. 7). To

further shorten the computational time, we set other control lines with 1° resolution (blue lines in Fig. 7) in the deeper ocean, where the global grid resolution is suggested in Tolman (2003). Note that the spatial locations of these two types of control lines are adjusted by constant testing to achieve a stable calculation and maximum benefit.

Step 3: generating the unstructured triangular multiscale grid. Once reasonable control lines and open boundaries (green lines in Fig. 7) are determined, a lot of triangles with different spatial sizes are quickly generated in SMS software. Some

adjustments to the poor mesh quality are suggested to ensure stable computation.



Now, the first version of the wave modeling framework using the unstructured triangular multiscale grid with the spatial resolution of 0.125°, 0.5°, and 1° in shallow, transitional, and deep water areas is finished ($WS_{multi3}^{utms}$ in Tab. 1). Fig. 7 shows that the spatial size of these meshes gradually and smoothly increases from coastal areas to deep oceans with the help of control lines. Currently, a tool named OceanMesh2D (Roberts et al., 2019) including a set of MATLAB functions is more
flexible and automated in making unstructured triangular multiscale grid systems. Then in the future, the grid optimization in the coastal area and the expansion of the regional grid to a global grid are very convenient and do not take much time.

## 4 Evaluation of wave simulations

### 4.1 Evaluation with different propagation schemes

Wave models describe the evolution of wave action density spectrum in the geographic space (including longitude and
latitude) and spectral space (including frequency and direction), dominated by source-sink terms. Since we only change the grid size in geographic space, here we will evaluate the performance of wave simulation using the unstructured triangular multiscale grid ($WS_{multi3}^{utms}$ in Tab. 1) in this space. There are four propagation schemes available in wave model WW3, including CRD-N, CRD-FCT, CRD-PSI, and implicit N. Please see Roland (2009) for more detailed descriptions. After 14-month numerical integration (from November 1, 2017, to December 31, 2018, UTC) using four numerical schemes
separately, $WS_{multi3}^{utms}$ can run stably. This indicates that it is feasible that wave energy can propagate smoothly and continuously on multiple meshes with different spatial resolutions. Simulation results are shown in Figure 8 and their computation time spent is listed in Table 3.

Fig. 8a displays SWH distributions of wave simulation using the propagation scheme CRD-N (the default scheme, first-order accuracy in time and space) in January 2018 (the month with the largest differences when wave simulation uses four
numerical schemes in 2018). There is a high correlation between the magnitude of wave height and wind intensity, for example, in the northern Pacific Ocean (the northern Indian Ocean and equatorial region), ocean surface waves have large (small) wave heights driven by strong (weak) wind. Figs. 8b-8d show the SWH differences between wave simulation using CRD-PSI, CRD-FCT, and implicit N schemes and that using the CRD-N scheme (Fig. 8a), respectively. The differences between wave simulation using nonlinear CRD-PSI and linear CRD-N schemes are relatively small (Fig. 8b) and the
calculation time spent of these two experiments is roughly the same (Tab. 3). Roland (2009) said that the CRD-PSI scheme is second order only in cross flow direction and is first order in longitudinal flow direction and time. There are obvious differences in Fig. 8c, especially in the complex topographic areas, because CRD-FCT has second-order accuracy in time and space, which also leads to its lowest calculation efficiency among the four schemes (Tab. 3). There are only slight differences in Fig. 8d because CRD-N and implicit N schemes both use a linear scheme. Although there are differences in
wave simulation results using four schemes, they can be negligible after verifying with observations at 9 available Chinese oceanic stations (Tab. 2). The wave parameters of the mean wave period (hereafter MWP) and mean wave direction (hereafter MWD) also have similar spatial distributions (not shown). On the whole, wave simulation with the explicit and





implicit schemes has similar simulation accuracy for Courant-Fredrichs-Levey (CFL) <1 (WW3DG, 2019). It should be noted that when wave simulation uses multiscale grid systems, it is better to extend the computing area outward by 3° to

reduce the influence of open boundaries on the concerned area, especially if the wave model uses the CRD-FCT scheme. Wave simulation results using the unstructured triangular grid with 0.125° resolutions in the whole domain ($WS^{ut}_{0.125}$ in Tab. 1) are regarded as a reference to evaluate the performance of $WS^{utms}_{multi3}$, whose comparison is listed in Tab. 3. Compared with $WS^{ut}_{0.125}$, simulation results of $WS^{utms}_{multi3}$ using four schemes are basically the same. Wave parameters of SWH and MWP both have small simulation differences and large correlation coefficients (hereafter CCs). The performance of MWD is

slightly worse than SWH and MWP but is acceptable. As we expected, wave simulation using a multiscale grid system has a high computational efficiency, saving more than 80% of computational time. This is consistent with the theoretical analysis in section 2.2.

The first three numerical schemes (including CRD-N, CRD-PSI, and CRD-FCT) are explicit schemes that are restrained by the CFL condition, which limits the nearshore resolution to 200 m in operational systems considering calculational costs.

The fourth scheme is an implicit scheme, refining coastal areas down to 10-50 m without CFL constraint. Although the implicit N scheme takes much computational time, it could be a good choice for accurate coastal wave simulation with the help of a new parallelization algorithm, domain decomposition (Abdolali et al. 2020). In the future, we can reasonably set the integral time step in the case of CFL>1, to further reduce calculation time costs. In this paper, since the current coastal resolution has not reached the order of 100 meters, the default scheme CRD-N will be adopted in the following study.

## 4.2 Evaluation of the influences of strong wind

The atmospheric wind is an important energy source for ocean surface waves (e.g., Roland and Ardhuin, 2014), then its seasonal characteristics will affect the evolution of wave states. Figure 9 shows the spatial distributions of SWH RMSDs in four seasons. Compared to the reference $WS^{ut}_{0.125}$, simulation differences of $WS^{utms}_{multi3}$ are very small in most ocean areas (less than 0.1 m) (left panels), such as equatorial region and northern Indian Ocean region. However, in the north of the

northern Pacific Ocean, there are obvious differences in all seasons, especially in winter (more than 0.15 m) shown in Fig. 9a. Similar visible differences also can be found in the west of the northern Pacific Ocean in the autumn (Figs. 9g and 9h). Wind distributions in this area (not shown) show that the equatorial region has very weak wind and the northern Indian Ocean is affected by monsoon but with moderate wind intensity. The north and west of the northern Pacific Ocean are affected by strong wind, westerly wind and tropical cyclones, respectively. We know that when the spatial resolution of wind forcing

and wave models is inconsistent, wind signals will be interpolated onto the wave model grid before model integration. Chen et al. (2018) compared the effect of coarse-grid wind forcing interpolating to fine-grid wave models on the wave energy. In this paper, we propose a hypothesis that if the wind is very strong and the wind direction changes rapidly, wind signals will be over-smoothed during the interpolation process (wind forcing with 0.25° resolutions and wave models with 1° resolution), resulting in poor wave simulation accuracy.





To confirm this hypothesis, we encrypt the unstructured triangular multiscale grid in the north of the northern Pacific Ocean for a preliminary test. As shown in Fig. 7, we divide the northern Pacific Ocean areas filled with grey (surrounded by a blue solid line) into two small areas named Area1 and Area2, delineated with a cyan dashed line (located at 27°N). Only the mesh resolution in Area1 is changed from 1° to 0.5°, and the mesh setting in Area2 remains the same as before. Now, the optimized unstructured triangular multiscale grid is generated. Using this grid, a similar numerical simulation (named $WS_{multi3(new)}^{utms}$ in Tab. 1) is done. We can see that its simulation differences in the northern Pacific Ocean are largely mitigated (less than 0.1 m) (right panels of Fig. 9) compared with $WS_{multi3}^{utms}$ (left panels of Fig. 9), while two experiments contain almost the same grid numbers (Tab. 1) and take almost the same computational time (in the following Fig. 13). While there are still some visible differences in the winter (Fig. 9b). This is because the integration of geographical, spectral advection terms and source terms is operated by using a fluctuation splitting scheme if wave model WW3 with the explicit scheme (here uses CRD-N). This will introduce splitting errors, especially when wind varies strongly in geographical space (Roland, 2009). If using the implicit N scheme, WW3 integrates the wave action equation directly without splitting error (Abdolali et al. 2020; Sikiric et al. 2018), then these differences are almost invisible at the current colorbar scale (not shown). Chen et al. (2018) mitigated the splitting error by using small time steps, but with little effect. Simulation differences of MWD and MWP are also alleviated. As their differences are small, the improvement is not as obvious as the SWH (not shown).

Different tropical cyclones vary greatly in time, space, and intensity, which will have important effects on wave simulation accuracy. The simulated SWH differences from wind sea and swell both have a high correlation with wind intensity in active typhon areas. The large differences often occur when the wind speed exceeds 50 m/s. As shown in Figs. 9e and 9g, the simulation difference locations overlap typhoon tracks (magenta lines) partially. Xu et al. (2017) stated that if the wind signal is not enriched from coarse grid to fine grid, only encrypting wave model resolution has little effect on wave simulation accuracy. Considering that typhoons cannot be accurately reproduced in the current coarse-resolution reanalysis atmosphere dataset ERA5 (0.25° resolutions, as shown in the following Figure 11b) (Hsiao et al., 2020), we preliminarily suggest that the grid resolution in whole active typhoon areas consistents with the spatial resolution of wind forcing to avoid missing wind signals. In the next paper, we will revise this long-duration reanalysis dataset using typhoon parameters, analyze the relationship between large differences and typhoon intensity, and then determine the specific area of multiscale grid encryption to further improve simulation efficiency.

In a word, in deep water areas, wave simulation using coarse-resolution grids can achieve the goal of enhancing computational efficiency without sacrificing simulation accuracy. According to the wind intensity, some suggestions are given for designing unstructured multiscale grid systems in these areas. (1) In active typhoon areas, we suggest preliminarily the spatial resolution of multiscale grid systems to be consistent with that of wind forcing to accurately capture the rapidly changing wind characteristics. (2) In the westerly zone, such as 30°N-60°N areas, the spatial resolution of multiscale grid systems should be twice coarser than that of wind forcing to avoid over-smoothing wind signals. (3) In moderate or weak





wind areas, the grid resolution of wave models could be 4 times coarser than that of wind forcing to shorten the computational time consumption.

## 4.3 Evaluation of influences of complex topography

With the advancement of HPC, high-resolution Earth system models have been widely developed. For example, Li et al. (2020) have developed three versions of coupled models in the Asia-Pacific area, of which the highest-resolution version is a 3 km atmosphere coupled with a 3 km ocean. However, the highest-resolution spatial resolution of the above designed unstructured triangular multiscale grid ($WS_{multi3}^{utms}$) in these areas is 0.125° (about 13 km), which is still hard to describe actual shorelines and complex topographic effects in coastal water areas. Here, supported by the fine dataset, we increase the grid resolution in coastal areas around the South China Sea (circled by a cyan solid box in Fig. 7) for further testing. The steps are as follows: (1) designing the shorelines with 0.0625° resolutions (about 7 km); (2) adjusting new control lines with 0.125° resolutions in suitable locations; (3) generating the new meshes in shallow water areas; and (4) replacing these meshes in the first version. Now, a finer unstructured triangular multiscale grid is finished (not shown). Then, a similar numerical experiment using it is done, named $WS_{multi4}^{utms}$ in Tab. 1.

Since the meshes are modified only in shallow water areas, we use observation data from the Chinese oceanic station BSG (marked with yellow stars in Figs. 6 and 7) to evaluate the above simulation results and the reference with a structured grid, named $WS_{0.0625}^{s}$ in Tab. 1. Figure 10 shows the scatter diagram of the observed and simulated SWHs at the valid observed time in four seasons of 2018. As described in section 3.1.2, wave simulation using a structured grid over-blocks wave energy at station BSG, resulting in the SWH underestimation. This situation is still not alleviated when the spatial resolution is increased from 0.125° (Fig. 6c) to 0.0625° (Figs. 10a, 10c, 10e, 10g). The $WS_{multi4}^{utms}$ without considering the island's blocking effect has a good performance (Figs. 10b, 10d, 10f, 10h). The SWH root mean square errors (RMSEs) are reduced by about 35% in every season. In terms of computational efficiency, $WS_{multi4}^{utms}$ (0.63 hours in Fig. 13) takes much less computational time than $WS_{0.0625}^{s}$ (33.79 hours in Fig. 3).

We further analyze the temporal evolution of observed and simulated wind speeds and SWHs in February and July 2018 (an example of winter and summer), respectively. Fig. 11 shows a good agreement between wind intensity and SWH magnitude from the observation, both plotted with black lines. When the wind is strong, the SWH is large, more obviously in July (Figs. 11b and 11d). Fig. 11 also shows the simulated SWHs driven by the same reanalysis wind, plotted with colored lines. It is noted that no observed wave data is available at this station because the spatial resolution of the ERA5 dataset is too coarse. Simulation results using the multiscale grid (red lines) and structured grid (blue lines) have a similar evolution but the former is closer to the observation (black lines), whether under low-moderate wind speeds (Fig. 11c) or high wind speeds as the typhoon passes through (typhoon Maria in Fig. 11d). Therefore, in shallow water areas, wave simulation using the unstructured multiscale grid can improve the description of complex shorelines and topography and enhance wave simulation precision. Furthermore, we see that wave simulation results are underestimated (e.g., Wu et al., 2020), especially





in the passing of tropical cyclones (e.g., Chen et al., 2018; Jiang et al., 2022). This indicates that the wind intensity from the ERA5 dataset needs to be corrected to obtain accurate wave states when typhoons occur. At the same time, we also learn that wave model WW3 using these two grids at other oceanic stations (DSN and ZLG, in Fig. 7) have similar underestimated behavior since the water depth at both stations is less than 10 m. Then it is urgent to enhance the simulated ability of wave models in coastal areas.

**4.4 Evaluation of the applicability**

Through the systematic tests above, we know that the WW3 wave model with a multiscale grid system is feasible and has a good performance in simulation accuracy and computational efficiency. Here, we will continue to test its applicability based on the previous section. Since there are a few triangles with 0.0625° resolutions in $WS_{multi4}^{utms}$, we still use $WS_{0.125}^{ut}$ as the reference for evaluation. Figure 12 shows that the two simulation results have negligible differences in 2018. In detail, the

RMSDs of SWHs, MWPs, and MWDs are all less than 0.1 meters, 0.23 seconds, and 32 degrees in Table 4, respectively. The CCs of SWHs and MWPs are around 0.99, and the MWD CCs are around 0.95. There is a slight but acceptable impact on MWD. A similar phenomenon can also be seen in Pallares et al. (2017), where the MWD is the most sensitive among these three variables when the used grid is changed. With the same computational resources (128 computing cores) simulating the same time length (31 days), $WS_{multi4}^{utms}$ takes 0.63 hours, saving about 70% of the calculational time compared

to the reference $WS_{0.125}^{ut}$ (2.04 hours), as shown in Fig. 13. Therefore, these results demonstrate that wave model WW3 using coarse-resolution grids in deep water areas has a negligible effect on wave simulation accuracy in the annual mean. It takes a small fraction of the computational time, compared with using the traditional structured grid with a fine resolution in the whole domain.

This efficient wave simulation using the unstructured triangular multiscale grid is beneficial to operational wave forecasting.

It can give faster warnings to minimize losses of coastal residents when catastrophic waves occur ($WS_{multi4}^{utms}$, 0.63 hours in Fig. 13), compared with wave simulation using the traditional structured grid with the same fine spatial resolution ($WS_{0.0625}^{s}$, 33.79 hours in Fig. 3). At the same time, it has fewer water points ($WS_{multi4}^{utms}$, 107, 317 in Tab. 1), 79% and 93% less than the reference ($WS_{0.125}^{ut}$, 521, 911) and the traditional simulation ($WS_{0.0625}^{s}$, 1, 632, 638), respectively. In atmosphere-ocean-wave coupled models, wave models using the unstructured triangular multiscale grid can reduce the integration time of the wave

component and shorten the flux-exchange time at the air-sea interface, finally enhancing the calculational efficiency of coupled systems. This suggests that this new wave modeling framework will accelerate the pace of high-resolution Earth system models including ocean surface waves as a fast-response physical process.

From the above detailed evaluation, we can conclude that a new wave modeling framework with an unstructured multiscale grid system can achieve the goals of less computational time consumption and better wave simulation precision, compared

with traditional wave simulations with uniform fine-gridding space. The applicability of this wave modeling framework is also verified by using two other wind forcings, including the ECMWF ERA-Interim dataset with 0.25° resolutions, and the



Climate Forecast System Version 2 (CFSR2) from the National Centers for Environmental Prediction (NCEP) with 0.2°
resolutions.

## 5 Summary and Discussions

This paper directly demonstrates that higher-resolution wave simulation can obtain more accurate wave state, but it requires
huge computational resources and has low computing efficiency. To deal with this situation, this paper designs a new wave
modeling framework with a multiscale system. It has the following advantages.

(1) Minimizing the number of fine grids. The wave dispersion relation regulating the wave modeling process shows that
ocean surface waves are insensitive to topographic effects in deep water areas. Then, the relationship between water depth

and half of the wavelength can be a criterion dividing shallow and deep water areas, which can decrease the number of fine
grids to the greatest extent. This way is more advantageous in Earth system models because it can shorten the added
computational time as much as possible when the ocean wave process is considered.

(2) Quantifying the match between grid resolution settings and wind signals. This paper gives some suggestions for
designing unstructured multiscale grid systems in deep water areas to avoid over-smoothing wind signals and enhance

computational efficiency. In active typhoon areas, westerly areas, and weak wind areas, the spatial resolution of multiscale
grid systems is suggested to be 1, 2, and 4 times coarser than that of wind forcing, respectively.

(3) Having similar accuracy using different spatial propagation schemes. This paper compares the performance of wave
simulation using four propagation schemes in geographic space, including CRD-N, CRD-PSI, CRD-FCT, and implicit N.
Four schemes have similar behavior in simulation accuracy, but the default CRD-N scheme takes the least computational

time.

(4) Achieving efficient and high-precision wave simulation. A series of experiment evaluations show that the designed wave
modeling framework can achieve the goals of enhancing wave simulation precision and saving computational costs. In deep
water areas, the wave model using the unstructured triangular multiscale grid ($WS_{multi3}^{utms}$) has very similar performance in
simulation accuracy but decreases 81% of the computational time consumption compared with using the unstructured

triangular grid ($WS_{0.125}^{ut}$). In shallow water areas, the wave model using the multiscale grid ($WS_{multi4}^{utms}$) can obtain more
accurate wave states and only takes 2% of the computational time compared with using the structured grid ($WS_{0.0625}^{s}$).

After establishing this powerful wave modeling framework, we will continue to conduct the following studies in the future.

(1) This framework can be constantly updated. For example, grid resolutions in coastal areas can be finer to better describe
complex topography, and other factors affecting the grid setting in deep water areas, especially in typhoon areas, should be

explored further.

(2) As HPC technology advances, the resolution of coastal grids will increase by up to hundreds of meters or even meters. It
is urgent to improve the simulated ability of wave models in coastal areas. In particular, the physical mechanism and
numerical scheme of wave models using multiscale grids should be improved.



(3) The spatial resolution between the available wind forcing and unstructured multiscale grids are often mismatched. The
linear interpolation method used in wave models may lose a lot of wind energy into wave models. Then a more reasonable
interpolation scheme should be explored.

(4) Ocean surface waves can be an important physical process to participate into Earth system models. Systematically
evaluating the contribution of ocean surface waves to the atmospheric planetary boundary layer and upper-ocean mixing will
be conducted. This will help us to deepen our understanding of the physical processes at the air-sea interface.

**Code and data availability**

The wave model WaveWatch III (WW3) used in this paper is from the Environmental Modeling Center (EMC), National
Oceanic and Atmospheric Administration (NOAA), and its source code can be downloaded from the website:
https://github.com/NOAA-EMC/WW3, last access: 9 February 2023. The Surface-water Modeling System software (SMS)
for making unstructured triangular (multiscale) grid systems is available from the website:
https://www.aquaveo.com/products, last access: 9 February 2023. The wind forcing is from the ERA5 dataset, European
Center for Medium-Range Weather Forecasts (ECMWF) (website:
https://cds.climate.copernicus.eu/cdsapp#!/dataset/reanalysis-era5-single-levels?tab=form, last access: 9 February 2023). The
shoreline data is from the NOAA GSHHS dataset (website: https://www.ngdc.noaa.gov/mgg/shorelines/data/gshhg/, last
access: 9 February 2023). The topography data comes from the NOAA ETOPO1 dataset (website:
https://www.ngdc.noaa.gov/mgg/global/global.html, last access: 9 February 2023). The observation data can be downloaded
from the National Marine Data Center, National Science & Technology Resource Sharing Service Platform of China
(website: http://mds.nmdis.org.cn/, last access: 9 February 2023). Finally, the data used to produce the figures in this paper
are available online (https:/doi.org/10.5281/zenodo.7587673, last access: 9 February 2023) or by sending a written request to
the corresponding author (Shaoqing Zhang, szhang@ouc.edu.cn).

**Author contribution**

Jiangyu Li designed the unstructured triangular multi-scale system, carried out all the experiments, and prepared the
manuscript with contributions from all co-authors. Shaoqing Zhang provided the scientific idea, analysed the results with
constructive discussions. Qingxiang Liu solved all the problems about wave models and checked carefully the words, figures
and tables of this manuscript. Xiaolin Yu and Zhiwei Zhang provided the test environment and intellectual discussion
necessary for the model design.





**Competing interests**

The authors declare that they have no conflict of interest.

**Acknowledgments**

Many thanks to Dr. Jianguo Li from Met Office, United Kingdom, for his constructive comments and suggestions. This
research is supported by the National Natural Science Foundation of China (41830964), the Shandong Province's "Taishan"
Scientist Project (ts201712017), and the Qingdao Postdoctoral Applied Research Project.

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



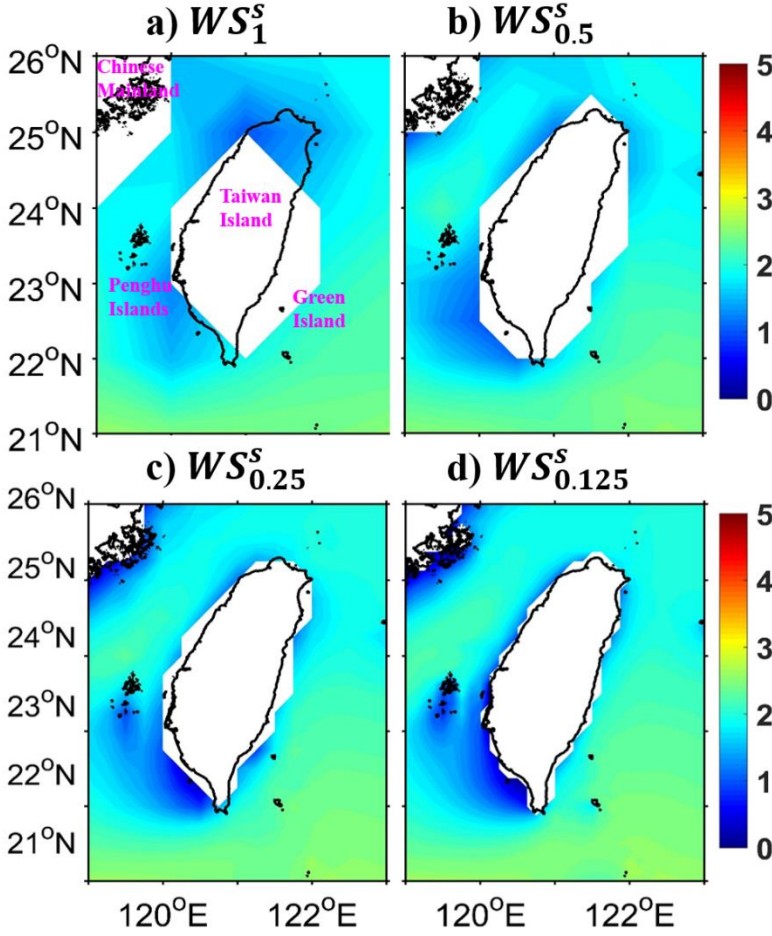

**Figure 1: Spatial distributions of significant wave heights (SWHs) from wave simulation (briefly as "WS") using a traditional structured grid system (briefly as "s" in the superscript) with a) 1°, b) 0.5°, c) 0.25°, and d) 0.125° model resolutions (denoted as the subscript) around Taiwan Island, China in January 2018, called $WS_1^s$, $WS_{0.5}^s$, $WS_{0.25}^s$, and $WS_{0.125}^s$ (see Tab. 1), respectively (unit: meter). The color-shaded and white indicate the ocean and land identified in wave model WW3 with different resolutions. The areas surrounded by black lines (from the GSHHS dataset) generally represent the real lands.**



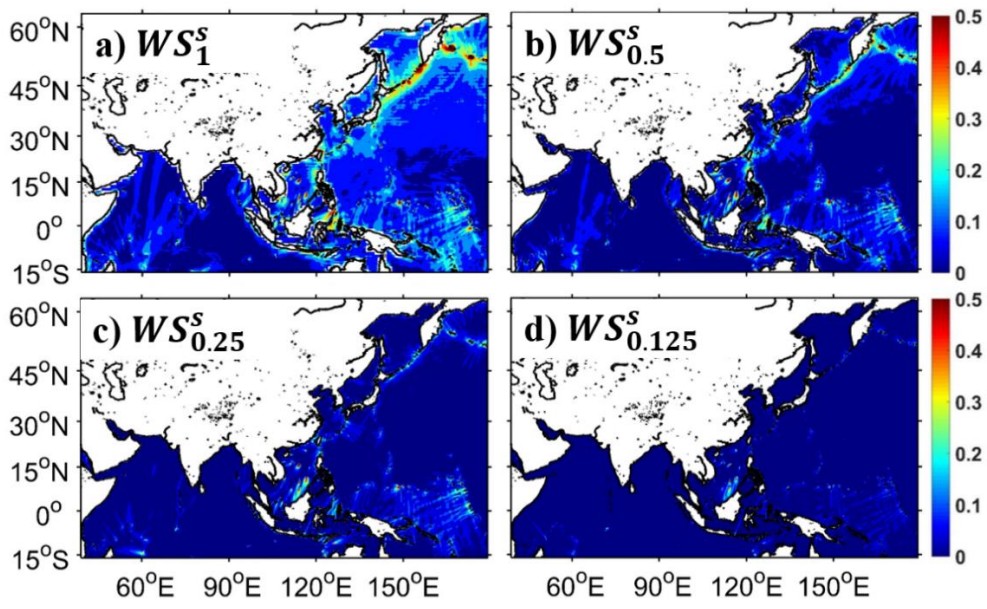

**Figure 2: Spatial distributions of SWH root mean square differences (RMSDs) from a) $WS_1^s$, b) $WS_{0.5}^s$, c) $WS_{0.25}^s$, and d) $WS_{0.125}^s$ around the Asia-Pacific area in January 2018 (unit: meter). The $WS_{0.0625}^s$ in Tab. 1 is considered as a reference to calculate four SWH RMSDs by linear interpolation.**

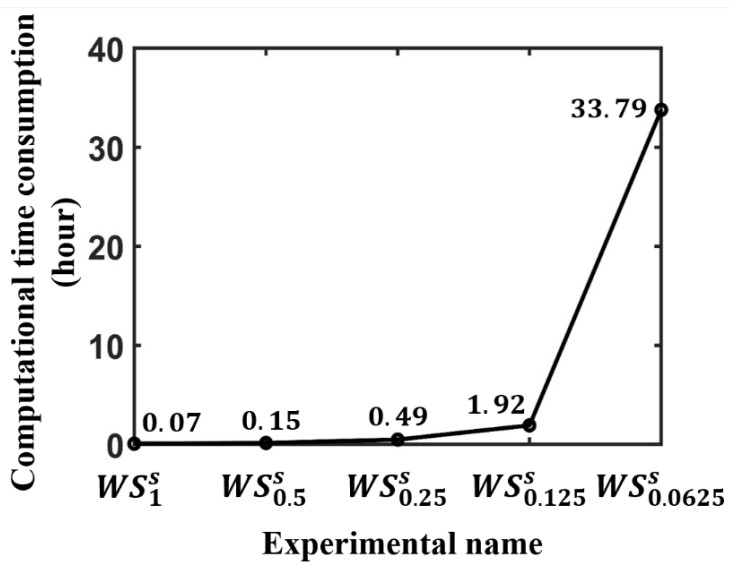

**Figure 3: The computational time consumption from $WS_1^s$, $WS_{0.5}^s$, $WS_{0.25}^s$, $WS_{0.125}^s$, and $WS_{0.0625}^s$ using the same computational resources (128 computing cores) to simulate one-month (January 2018) wave states around the Asia-Pacific area. The specific time consumption is listed at the corresponding position.**



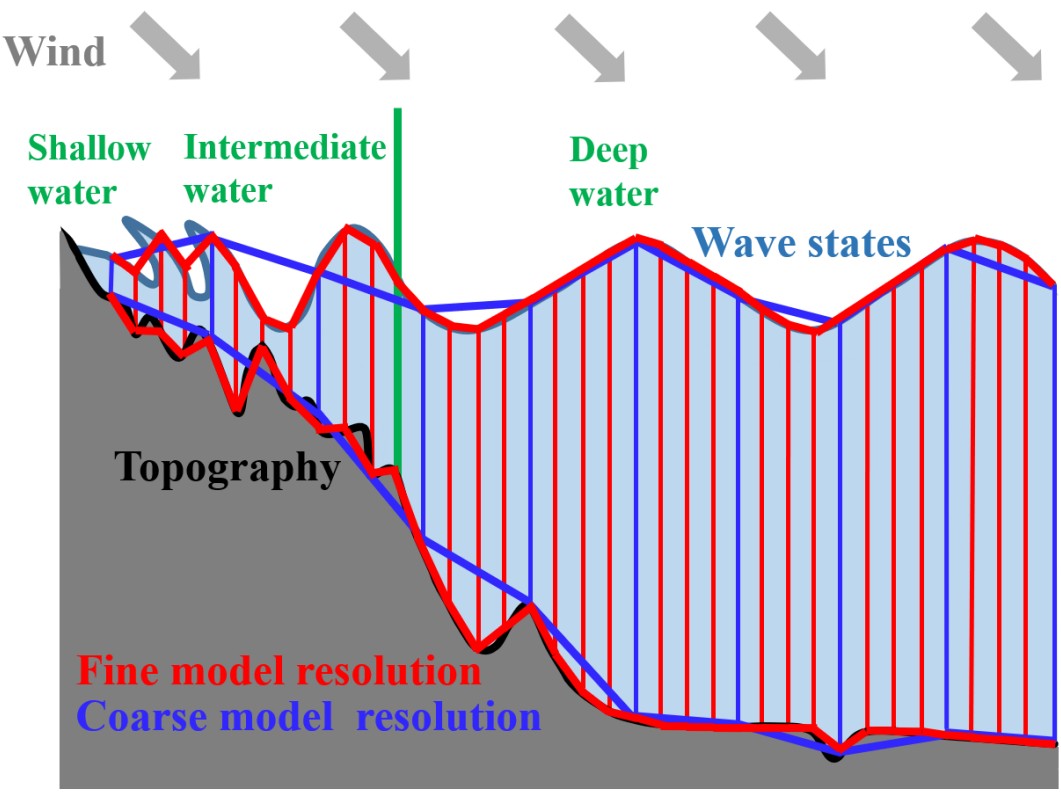

**Figure 4: A schematic diagram of wave models describing complex topographic features (grey fill) and simulating wave states (navy-blue lines) using fine (red lines) and coarse (blue lines) model resolutions in shallow, intermediate, and deep water areas (left and right of the green vertical bar). The black and navy-blue lines represent the actual land-ocean boundary and wave states, which are described with the thick red and blue lines in wave models. Note that this figure does not represent the actual wave modeling process and the spatial scale of ocean surface waves.**





**Figure 5: Spatial distributions of SWH differences from $WS_1^s$ (a, e, i, m), $WS_{0.5}^s$ (b, f, j, n), $WS_{0.25}^s$ (c, g, k, o), and $WS_{0.125}^s$ (d, h, l, p) around the South China Sea at 01:00, 06:00, 12:00 UTC, November 1, 2017 (the first, second, third row, named T1, T6, T12), and 00:00 UTC, November 2, 2017 (the fourth row, named T24) (note that the wave states of all experiments at 00:00 UTC, November 1, 2017, are resting) (unit: meter). The Zhongsha Islands are circled by dashed boxes in the first column and the last row. The $WS_{0.0625}^s$ in Tab. 1 is considered as a reference to calculate SWH differences by linear interpolation (interpolated results minus the reference).**



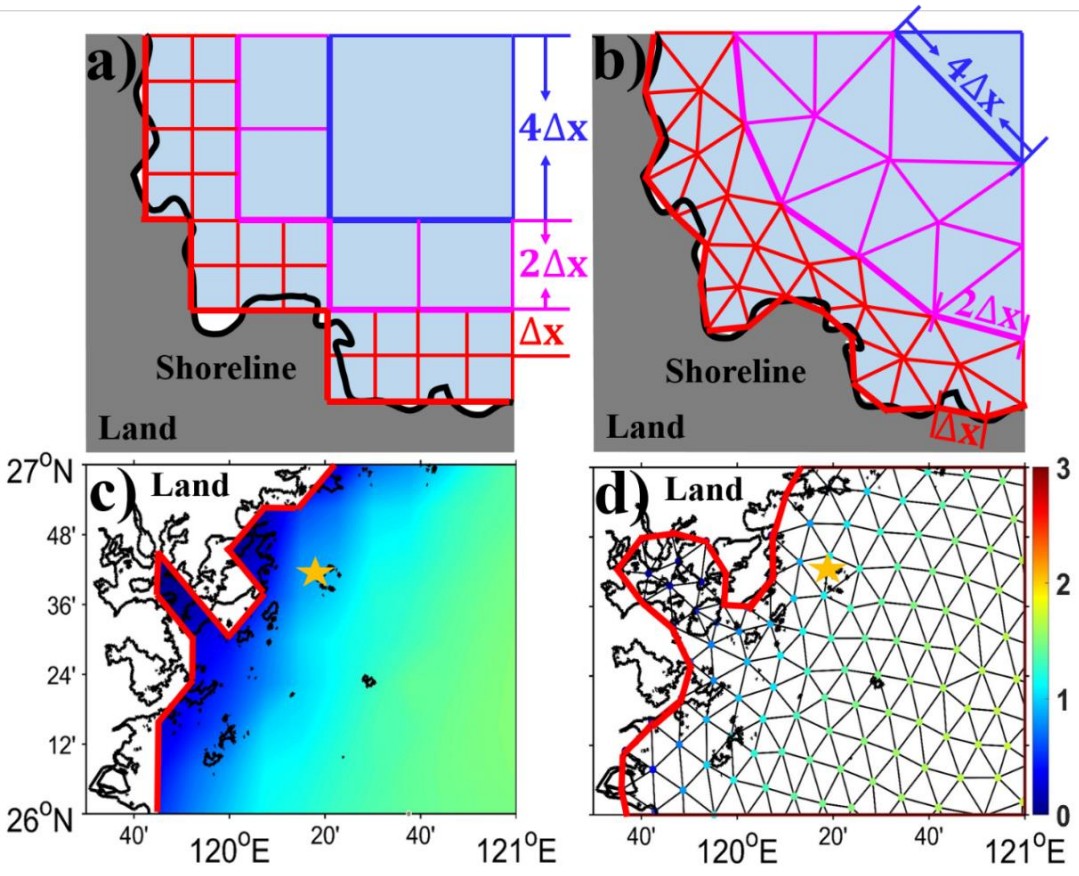

**Figure 6: A diagram of unstructured a) rectangular and b) triangular multiscale grid systems with Δx, 2Δx, and 4Δx spatial resolutions in shallow, transitional, and deep water areas marked with red, magenta, and blue lines. Spatial distributions of SWHs are from wave simulation using c) traditional structured grid and d) unstructured triangular grid both with a fine resolution (named $WS^{s}_{0.125}$ and $WS^{ut}_{0.125}$ in Tab. 1) in July 2018 (unit: meter). The Chinese oceanic station named BSG is located at (120.3°E, 26.7°N) marked with yellow stars in c) and d). The thick black and red lines are actual and described land-ocean boundaries in four panels.**



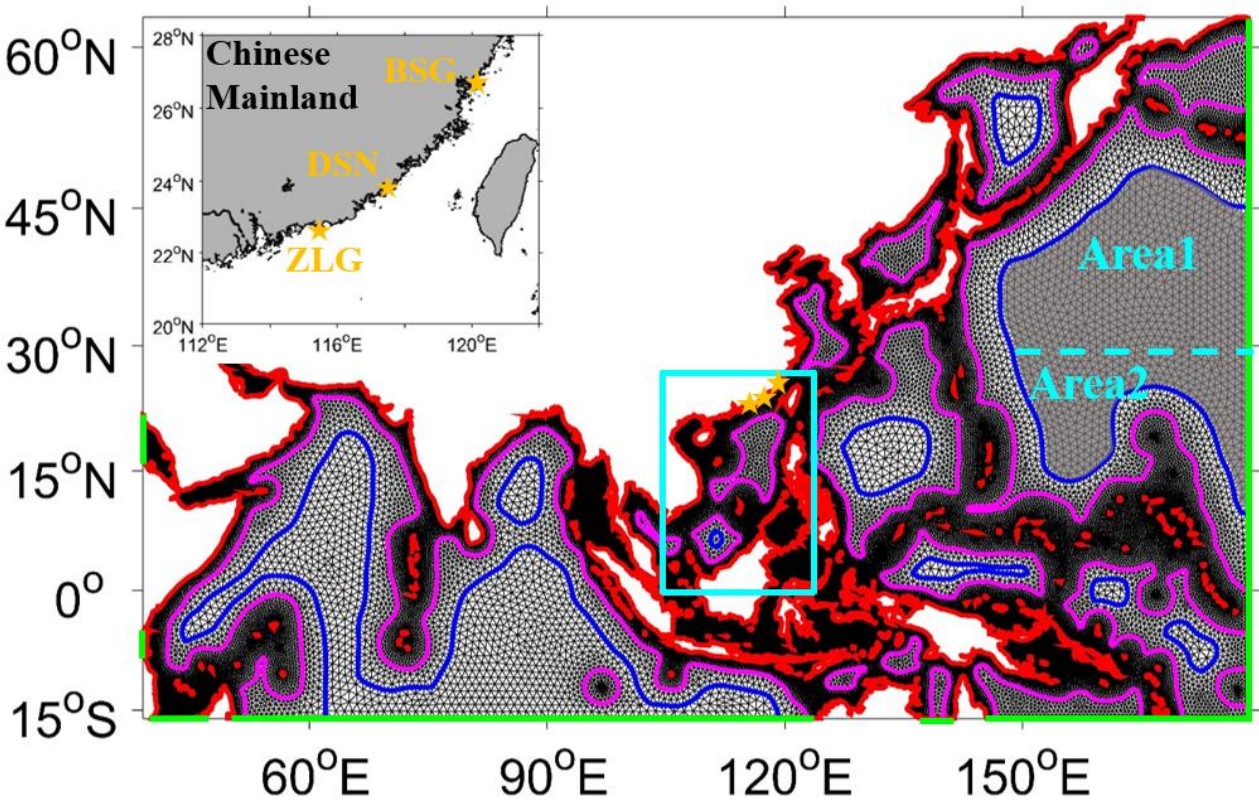

**Figure 7: The spatial distribution of the new wave modeling framework using an unstructured triangular multiscale grid system. Grid resolutions vary from 0.125° in shallow water areas to 0.5° in transitional water areas and then to 1° in deep water areas, with the help of the shorelines (red) and two types of control lines (magenta and blue), named $WS_{multi3}^{utms}$ in Tab. 1. The green lines represent spatial locations of the open boundary. The Chinese oceanic stations named BSG (same station in Fig. 6), DSN, and ZLG are marked with yellow stars, and the top-left panel is a clearer display. In the following section 4, this framework will be further developed in two areas. The first is the northern Pacific Ocean area with a grey fill (surrounded by a blue line) ($WS_{multi3(new)}^{utms}$). The second is around the South China Sea area circled by a cyan solid box ($WS_{multi4}^{utms}$). Please, see the corresponding part for details.**



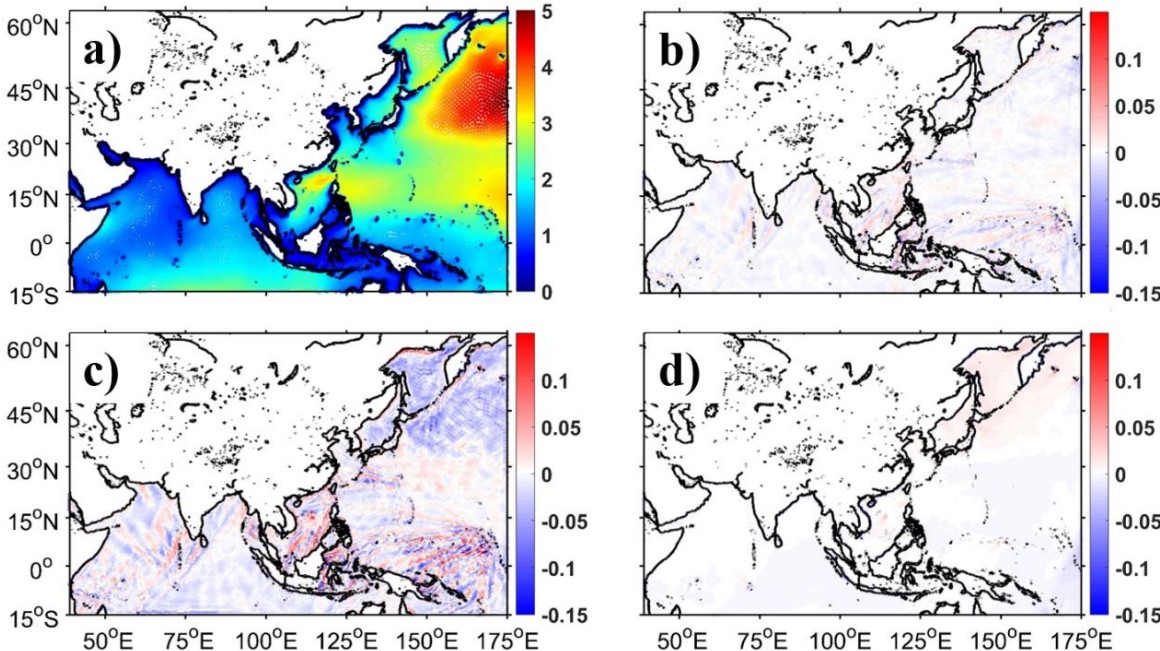

**Figure 8: a) Spatial distributions of SWH from $WS_{multi3}^{utms}$ using CRD-N propagation scheme in January 2018 (unit: meter). b-d) Spatial distributions of SWH differences from $WS_{multi3}^{utms}$ using CRD-PSI, CRD-FCT, implicit N propagation schemes minus that using CRD-N scheme (Fig. 8a), respectively (unit: meter).**




**Figure 9: Spatial distributions of SWH RMSDs from $WS^{utms}_{multi3}$ (a, c, e, g) and $WS^{utms}_{multi3(new)}$ (b, d, f, h) in the winter (a, b), spring (c, d), summer (e, f), and autumn (g, h) of 2018 (unit: meter). The reference $WS^{ut}_{0.125}$ is used to calculate SWH RMSDs by linear interpolation. The magenta lines in panels e and g are best tracks of some typhoons, which overlap the locations of large differences partially.**



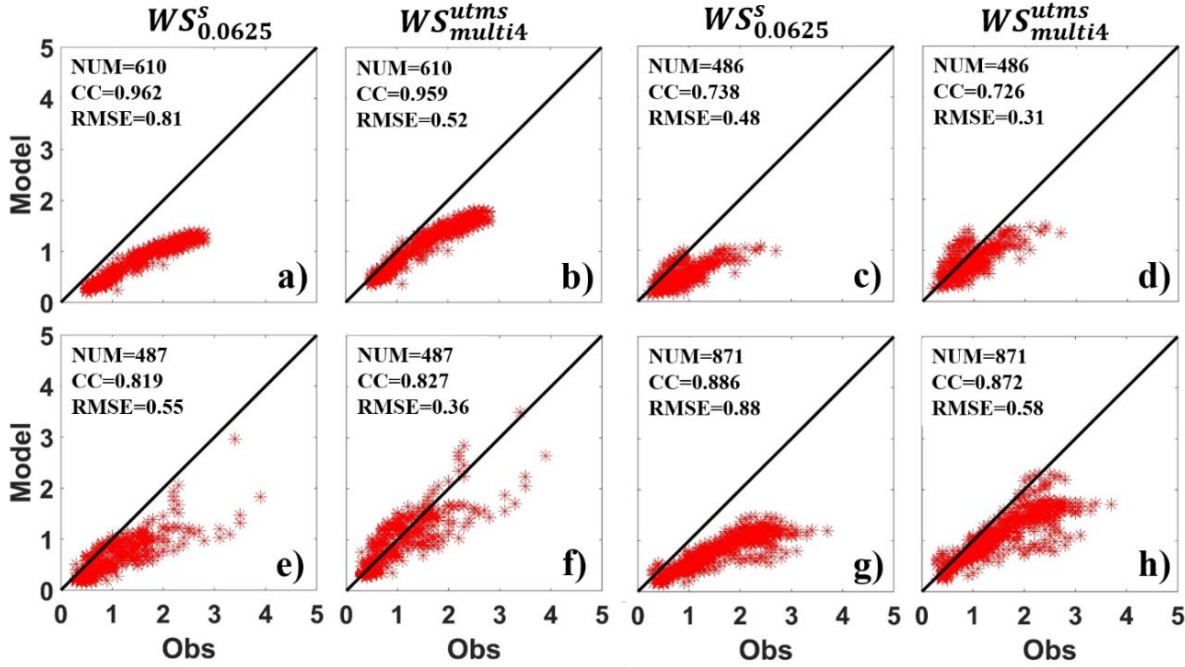

**Figure 10: Scattered distributions of SWHs from $WS^s_{0.0625}$ (a, c, e, g) and $WS^{utms}_{multi4}$ (b, d, f, h) at the observational station BSG (marked with yellow stars in Figs. 6 and 7) in the winter (a, b), spring (c, d), summer (e, f), and autumn (g, h) of 2018 (unit: meter). The black lines in every panel indicate the best fit between wave simulation results (the vertical axis) and observations (the horizontal axis). The number of valid observations and the calculated SWH root mean square errors (RMSEs) and CCs are listed in the upper-left corner of every panel.**




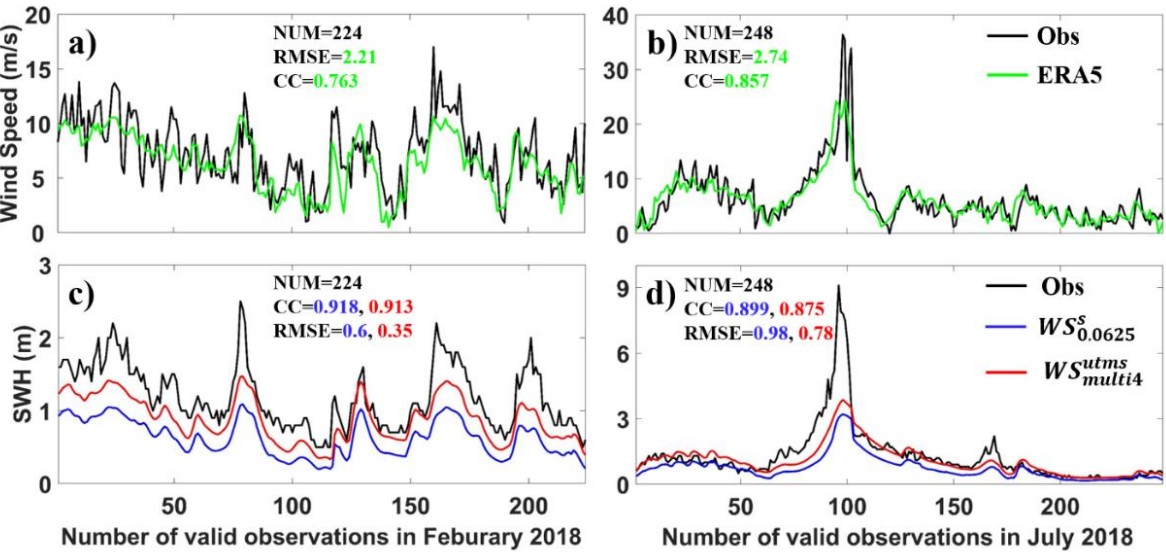

**Figure 11: Time series of wind speeds (a, b) and SWHs (c, d) at the BSG observation station in February (a, c) and July (b, d), 2018. Wind speeds and SWHs observed are plotted with black lines. The wind forcing is from the reanalysis dataset ERA5 plotted with green lines in a) and b). The simulated SWHs from $WS_{0.0625}^{s}$ and $WS_{multi4}^{utms}$ are plotted with blue and red lines in c) and d), respectively.**

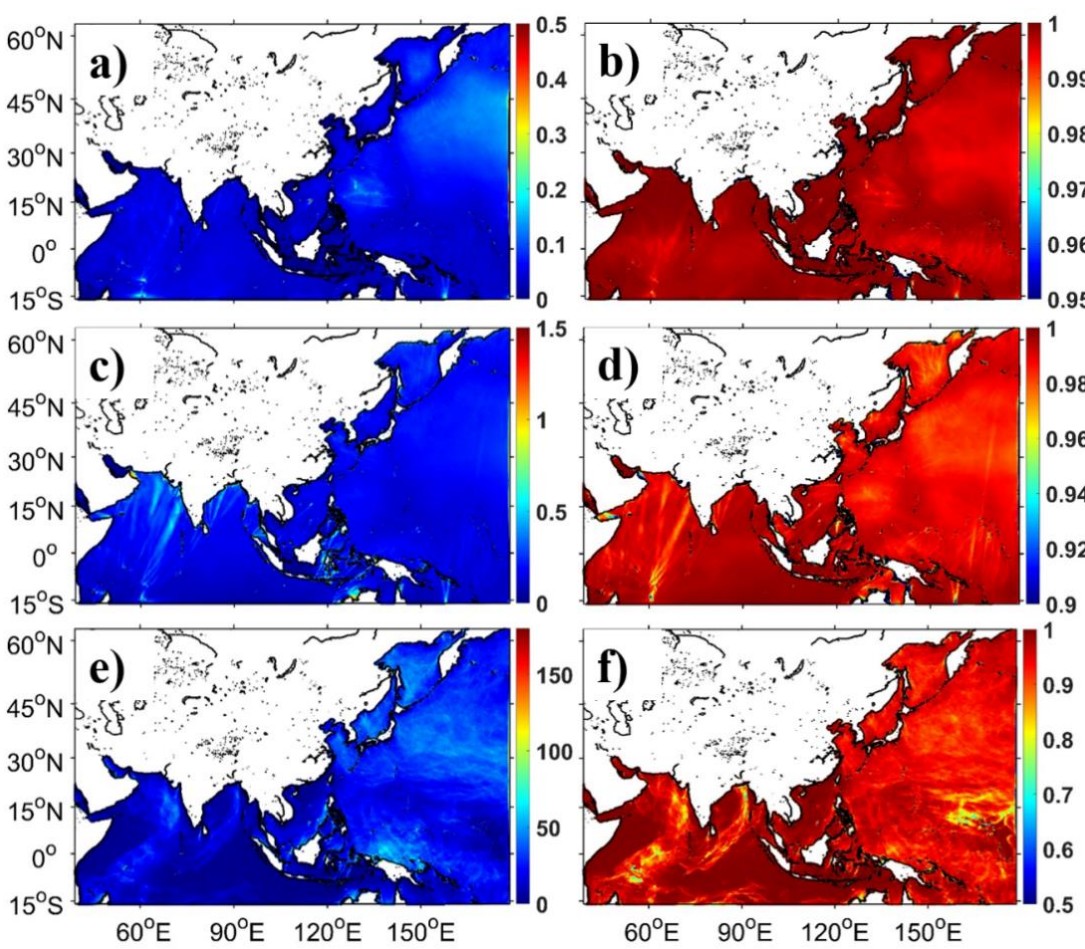

**Figure 12: Spatial distributions of RMSDs (a, c, e) and CCs (b, d, f) of SWHs (a, b), MWPs (c, d), and MWDs (e, f) from $WS_{multi4}^{utms}$ in 2018 (unit of panel a/c/e: meter/second/degree). The $WS_{0.125}^{ut}$ in Tab. 1 is considered as a reference to calculate the RMSDs and CCs by linear interpolation.**



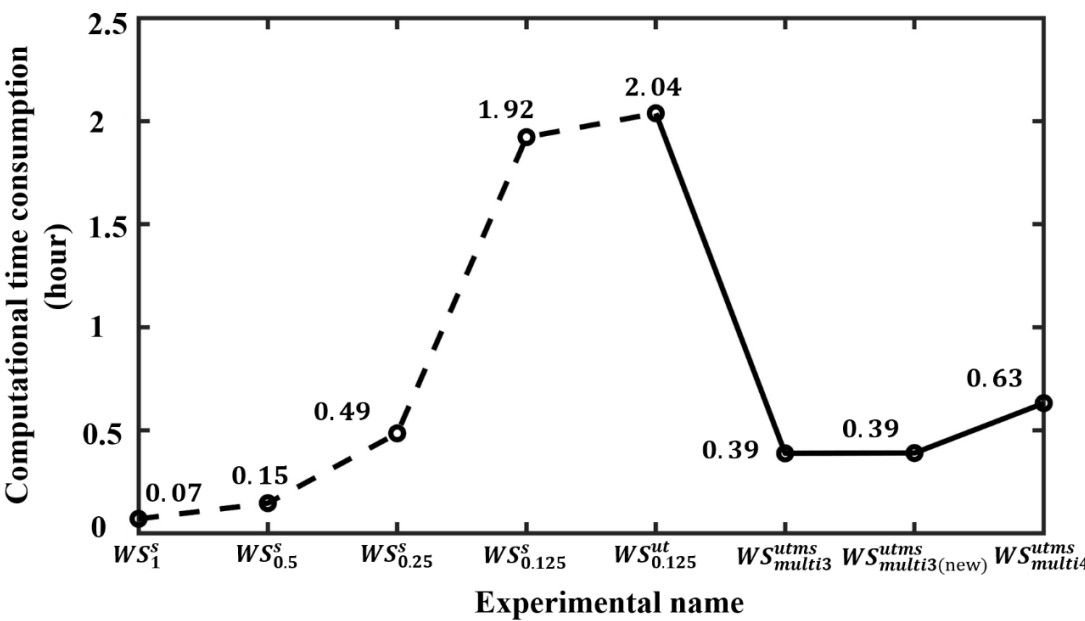


**Figure 13: The computational time consumption from wave simulation with unstructured triangular (multiscale) grid systems (solid lines) under the same computational condition. The computational time consumption from Fig. 3 is plotted here with dashed lines for comparison.**



**Table 1: Design of wave simulation experiments with different grid systems and model resolutions.**

| The name of the experiments | Grid types | Model resolutions | Numbers of water points (or nodes) | The role of experiments |
|---|---|---|---|---|
| $WS_1^s$ | Structured grid | 1°×1° | 6, 454 | The performance of wave simulation with different model resolutions |
| $WS_{0.5}^s$ | | 0.5°×0.5° | 25, 626 | |
| $WS_{0.25}^s$ | | 0.25°×0.25° | 102, 325 | |
| $WS_{0.125}^s$ | | 0.125°×0.125° | 408, 511 | |
| $WS_{0.0625}^s$ | | 0.0625°×0.0625° | 1, 632, 638 | The reference |
| $WS_{0.125}^{ut}$ | Unstructured triangular grid | 0.125° | 521, 911 | |
| $WS_{multi3}^{utms}$ | Unstructured triangular multiscale grid | 0.125°, 0.5°, and 1° in shallow, transitional, and deep water areas | 90, 652 | The performance of wave simulation in strong wind areas |
| $WS_{multi3(new)}^{utms}$ | | 0.125°, 0.5°, and 1° in shallow, transitional, and deep water areas (slight changes in the northern Pacific Ocean area) | 91, 472 | |
| $WS_{multi4}^{utms}$ | | 0.0625°, 0.125°, 0.5°, and 1° in coastal, shallow, transitional, and deep water areas (slight changes around the South China Sea area) | 107, 317 | The performance of wave simulation in complex topography areas |

**Note: to reduce the uncertainty, the maximum global time step, maximum CFL time step for geographic and spectral space, and minimum source-sink term time step in all experiments are the same, which are 900s, 90s, 300s, and 10s, respectively.**





**Table 2: the formation of Chinese oceanic observation stations used in this paper.**

| Station name | Longitude (°E) | Latitude (°N) | Water depth (m) | Data available in 2018 |
|:---:|:---:|:---:|:---:|:---:|
| XMD | 120.4 | 36.0 | 19.4 | Jan. – Dec. |
| XCS | 122.7 | 39.2 | 16.7 | Jan. – Dec. |
| NJI | 121.1 | 27.5 | 16 | Jan. – Dec. |
| BSG | 120.3 | 26.7 | 10.4 | Jan. – Dec. |
| LHT | 121.7 | 38.9 | 9.5 | Jan. – Mar., May – Dec. |
| ZLG | 115.6 | 22.7 | 8.3 | May, Jul. – Sep. |
| DCN | 121.9 | 28.5 | 5.7 | Jan., Feb., May – Dec. |
| LYG | 119.4 | 34.8 | 4.7 | Jan. – Dec. |
| DSN | 117.5 | 23.8 | 1.7 | Feb., Mar., May, Jul. – Dec. |


**Table 3: The performance of $WS_{multi3}^{utms}$ and the reference $WS_{0.125}^{ut}$ both using four propagation schemes in January 2018.**

| Propagation scheme | SWH | | MWP | | MWD | | Computational time (hour) | | |
|:---:|:---:|:---:|:---:|:---:|:---:|:---:|:---:|:---:|:---:|
| | RMSD (m) | CC | RMSD (s) | CC | RMSD (°) | CC | $WS_{0.125}^{ut}$ | $WS_{multi3}^{utms}$ | Improved (%) |
| CRD-N | 0.06 | 0.991 | 0.18 | 0.984 | 23.72 | 0.927 | 2.04 | 0.39 | 81% |
| CRD-PSI | 0.08 | 0.986 | 0.2 | 0.979 | 24.8 | 0.92 | 2.11 | 0.4 | 81% |
| CRD-FCT | 0.08 | 0.986 | 0.21 | 0.978 | 25.22 | 0.915 | 4.3 | 0.74 | 83% |
| Implicit N | 0.07 | 0.988 | 0.18 | 0.982 | 23.64 | 0.928 | 3.84 | 0.67 | 83% |

**Note: simulation results of $WS_{multi3}^{utms}$ are interpolated onto the reference grid to calculate the RMSDs and correlation coefficients (CCs) of SWH, mean wave period (MWP), and mean wave direction (MWD), respectively.**

**Table 4: The RMSDs and CCs statistics of SWHs, MWPs, and MWDs from $WS_{multi4}^{utms}$ compared with the reference $WS_{0.125}^{ut}$ in 2018.**

| Seasons (months) | SWH | | MWP | | MWD | |
|:---:|:---:|:---:|:---:|:---:|:---:|:---:|
| | RMSD (m) | CC | RMSD (s) | CC | RMSD (°) | CC |
| Winter (DJF) | 0.09 | 0.996 | 0.21 | 0.993 | 31.14 | 0.957 |
| Spring (MAM) | 0.07 | 0.997 | 0.21 | 0.994 | 21.54 | 0.964 |
| Summer (JJA) | 0.06 | 0.998 | 0.19 | 0.995 | 17.81 | 0.962 |
| Autumn (SON) | 0.08 | 0.996 | 0.22 | 0.993 | 26.87 | 0.948 |
| Annual mean | 0.08 | 0.997 | 0.21 | 0.994 | 24.34 | 0.958 |