# Peer review of "Design and Evaluation of an Efficient High-Precision Ocean Surface Wave Model with a Multiscale Grid System (MSG\_Wav1.0)"

_Geoscientific Model Development, 2023_

## Author Response (AR1)

**Interactive comments on "Design and Evaluation of an Efficient High-Precision Ocean Surface Wave Model with a Multiscale Grid System (MSG_Wav1.0)" from anonymous referee #1**

This paper illustrates a significant work in high-resolution wave model development. The multiscale gridding framework shows it can mitigate the high computing cost while keeping a good performance in coastal regions. The reasoning is clear supported by evidence. I believe this technique will make a significant contribution to HRM development. I only have a few minor comments/suggestions. See them below.

RE: Thank you very much for your endorsement and excellent suggestions, which give us more thinking, especially in the applicability of wave modeling with a multiscale grid in other wave models and Earth system models. In this revision, we (1) have added more discussions about the applicability of the wave modeling framework with a multiscale system; (2) have moved several pieces of content to a more suitable place and further optimized them; (3) have revised inappropriate language description, such as: using objective words instead of subjective words; giving clearer descriptions for vague expressions.

1. Line 103: 'acceptable' is a subjective word. Suggest removing it or changing to an objective word.

   RE: Thanks for your good suggestion. We have changed this inaccurate description. Please see lines 114-115.

2. Line 207-208: what is 'a reasonable range'? Better to say it explicitly.

   RE: Thanks for your kind reminder. We have given the specific range used in this manuscript. Please see line 228.

3. Lines 259-261: is it better to move this to discussion section?

   RE: Thanks for your good suggestion. We have moved this part to Section 5 and further optimized it. Please see lines 541-545.

4. Line 295: again, 'acceptable' is a subjective word.

   RE: Thanks for your good suggestion. We have changed this improper expression. Please see lines 326-328.

5. Paragraph with Line 300: any better to move it to discussion section?

RE: Thanks for your good advice. We have moved this paragraph to Section 5 and further optimized it. Please see lines 546-550.

6. Paragraph with Line 325: is it true that WS_multi3 and WS_multi3(new) have the same grid number? Resolution in Area1 is higher in WS_multi3(new), meaning a (slightly?) larger number of grid than in WS_multi3.

RE: Thanks for your kind reminder. We have changed this inaccurate description. Please see lines 380-382.

7. Lines 414-417: there are no analysis on era-i and CFSR2 forced wave simulations in this paper. Remove this or put as a discussion point?

RE: Thanks for your kind reminder. We have moved it to Section 5 as a discussion point. Please see lines 561-564.

8. Line ~440: WS_multi4 has the finest resolution only in SCS, which is a small proportion in the coastal regions of the whole domain. If the gridding strategy in WS_multi4 is applied to the whole domain, the computing demand could increase significantly. It deserves a clarification for this point, otherwise it may cause a misleading.

RE: Thanks for your excellent suggestion. We have added a more detailed description to make it clearer. Please see lines 520-529.

9. Future work around line 445: the $1^{st}$ and $2^{nd}$ points are not distinctive as some thoughts are mixed up. Suggest rephrasing these two paragraphs.

RE: Thanks for your helpful suggestion. We have rewritten these two paragraphs to make their meanings independent. Please see lines 541-550.

10. Future work around line 452: it's worth mentioning in a coupling context, whether the feedback of waves to the atmosphere and ocean is sensitive to wave multiscale resolution. For example, one may assume that a coarser resolution wave model produces less wave effect on atm and ocean. Does this mean inhomogeneity in wave feedbacks in this multiscale framework?

RE: Thanks for your thoughtful advice. We have refined this discussion about the application of this wave modeling framework to Earth system models in Section 5. Please see lines 572-578.

11. Summary and discussions: Are the conclusions and recommendations in this paper applicable to other wave models as well e.g. SWAN and WAM? It's worth having a thought.

RE: Thanks for your excellent suggestion. We have added a discussion about the application of this multiscale grid to other wave models in Section 5. Please see lines 567-571.

**Interactive comments on "Design and Evaluation of an Efficient High-Precision Ocean Surface Wave Model with a Multiscale Grid System (MSG_Wav1.0)" from anonymous referee #2**

The study of wave numerical simulation is of great value to both practical application and scientific research. However, development of a high-precision wave model with highly-efficient computation is very difficult. This paper studies the effect of model resolution on wave simulation in shallow and deep water areas and then designs a reasonable unstructured multiscale grid system to balance simulation accuracy and computational efficiency. The novelty and application value are high. I recommend publication after minor revision. See below for specific points.

RE: Thanks for your constructive comments, which have helped us greatly to evaluate the performance of wave modeling with a multiscale grid system more clearly. In this revision, we have modified the manuscript in strict accordance with your comments and suggestions, mainly including four aspects: (1) adding a clearer description of the design and operation of the control and reference experiments; (2) adding more wind information in some figures; (3) giving more reasonable description in Section Evaluation of wave simulations; (4) giving more detailed expression about some unclear or inaccurate sentences and method choices.

1. Line 47: about the sentence "It is inconvenient for high-precision operational wave forecasting …", operational wave forecasting systems pay attention to both simulation accuracy and calculation efficiency.

   RE: Thanks for your good comment. We have rewritten this sentence to get a better expression. Please see lines 47-50.

2. Line 55: about the sentence "… we design a new … based on a multiscale grid system", you can simply describe this "system" to give readers a general impression.

   RE: Thanks for your good advice. We have added the corresponding description to give the reader a general impression. Please see line 62.

3. Lines 92-93: which way do you use in this paper? Being approximated with obstruction grids? Being parameterized with a source term? Or something else?

   RE: Thanks for your kind reminder. We have mentioned the way we used in this manuscript. Please see line 105.

4. Lines 103-105: please give a more detailed explanation about the dramatic increase in computing time.

RE: Thanks for your excellent suggestion. We have added a more detailed explanation of this situation. Please see lines 116-121.

5. Lines 109-111: about the sentence "In the future, … the atmosphere-ocean coupled models", I feel that it is a half sentence, the meaning of the expression is incomplete.

   RE: Thanks for your good advice. We have added the corresponding content to make the meaning clearer. Please see line 128.

6. Lines 153-154: about the sentence "As we expected … gradually decrease", the increase of model resolution not only improves the representativeness of topographic features but also improves the response to local wind.

   RE: Thanks for your great comment. We have added the corresponding content to get a more proper description. Please see lines 173-174.

7. Lines 211-212: about Figs 6c and 6d, is it only a small area drawn for clarity? In Tab. 2, some of the nine stations are located outside this area.

   RE: Thanks for your kind reminder. We have mentioned this point to give readers a clearer understanding. Please see lines 232-233.

8. Line 247: about the sentence "… shallow and deep water areas …", you can note their definition from section 2.3 to avoid confusion with the classical definition.

   RE: Thanks for your kind advice. We have added the definition used in this manuscript to avoid confusion with the classical definition. Please see lines 268-269.

9. Line 255: how to identify and deal with poor meshes? Does SMS have this function?

   RE: Thanks for your helpful comment. We have given a more detailed description of how to identify and deal with poor meshes using the SMS tool, and the identification criteria used in this manuscript. Please see lines 276-280.

10. In section 4.1, about the description of "first-order accuracy" and "second-order accuracy", the word "precision" is more commonly used than "accuracy".

    RE: Thanks for your kind reminder. We have replaced this inaccurate word. Please see lines 299 and 309.

11. Lines 288-290: about the sentence "It should be noted that…especially if the wave model uses the CRD-FCT scheme", is it because the CRD-FCT scheme has a two-order precision?

RE: Thanks for your thoughtful comment. We have added the corresponding description. Please see lines 320-321.

12. Line 291: does the reference also use four schemes or only use the CRD-N scheme?

   RE: Thanks for your kind reminder. We have added a clearer description of the reference design to make the comparison between the reference and control experiment easier to follow. Please see lines 323-325.

13. Lines 300-302: about the sentence "Although the implicit N scheme … domain decomposition", there is an ambiguity here. Does the new algorithm help save calculation time or improve simulation accuracy?

   RE: Thanks for your thoughtful comment. We have rewritten the corresponding description to make the meaning clearer and moved it to Section 5. Please see lines 546-550.

14. Lines 315-316: I don't know the intent of the sentence "Chen et al. (2018) …".

   RE: Thanks for your helpful comment. We have rewritten the study about the influence of a smoothing wind on ocean surface waves from Chen et al. (2018). Please see lines 355-357.

15. Lines 321-323: about the sentence "As shown in Fig.7, … remains the same as before", leaving everything else the same and only dealing with the filled area?

   RE: Thanks for your kind reminder. We have added a detailed description of generating a new multiscale grid. Please see line 362.

16. Lines 328-330: about the sentence "While there are still … with the explicit scheme", the spatial resolution of wind forcing and wave models is inconsistent, which also is an intuitive reason when the wind speed is very high and the wind direction changes quickly.

   RE: Thanks for your excellent suggestion. We have rewritten the reasons for visible simulation differences in boreal winter in the North Pacific Ocean. Please see lines 369-378.

17. Line 337: you have better to delete "from wind sea and swell both" because you don't mention them in the following.

   RE: Thanks for your good advice. We have deleted these words. Please see line 385.

18. Lines 378-379: is this a mistake? "no observed wave data in ERA5 dataset"?

RE: Thanks for your kind reminder. We have changed this inaccurate description. Please see lines 438-440.

19. Lines 384-389: I don't understand these sentences "Furthermore, … in coastal areas", it seems to have a gap with the previous meaning.

RE: Thanks for your helpful comment. We have rewritten this part and moved some of it to a more suitable place. Please see lines 389-391 and 446-452.

20. In section 5: Recommend authors to add some discussions on how to extend this grid to global domain.

RE: Thanks for your thoughtful advice. We have added some discussions about how to optimize multiscale grids (including the extension of a regional grid to a global domain) in Section 5. Please see lines 541-545.